# Dome-Shaped Macula versus Ridge-Shaped Macula Eyes in High Myopia Based on the 12-line Radial Optical Coherence Tomography Scan Pattern. Differences in Clinical Features

**DOI:** 10.3390/diagnostics11101864

**Published:** 2021-10-11

**Authors:** María García-Zamora, Ignacio Flores-Moreno, Jorge Ruiz-Medrano, Rocío Vega-González, Mariluz Puertas, Elena Almazán-Alonso, Lucía González-Buendía, José M. Ruiz-Moreno

**Affiliations:** 1Department of Ophthalmology, Puerta de Hierro-Majadahonda University Hospital, 28222 Madrid, Spain; i_floresmoreno@hotmail.com (I.F.-M.); jorge.ruizmedrano@gmail.com (J.R.-M.); rociovgonzalez@hotmail.com (R.V.-G.); mariluzpuertas@gmail.com (M.P.); elena.almazanalonso@gmail.com (E.A.-A.); luciaglezbuendia@gmail.com (L.G.-B.); josemaria.ruiz@uclm.es (J.M.R.-M.); 2Department of Ophthalmology, Castilla La Mancha University, 02001 Albacete, Spain; 3Red Temática de Investigación Cooperativa en Salud: “Prevención, Detección Precoz y Tratamiento de la Patología Ocular Prevalente, Degenerativa y Crónica” (RD16/0008/0021), Spanish Ministry of Health, Instituto de Salud Carlos III, 28029 Madrid, Spain; 4Miranza Coorporation, 28035 Madrid, Spain

**Keywords:** retina, myopia, dome-shaped macula, ridge-shaped macula

## Abstract

Purpose: To study clinical features in patients with ridge-shaped macula (RSM) compared with those with dome-shaped macula (DSM) having been previously classified by the number of swept-source optical coherence tomography (SS-OCT) radial scans affected. Methods: Retrospective observational study including 49 highly myopic eyes from 31 patients who underwent SS-OCT. DSM eyes were defined as those that showed a complete round inward convexity in all their axes, presenting an inward convexity ≥50 µm in the 12-line radial OCT scans. Eyes that did not meet this criterion and had at least one flat radial scan were grouped into the RSM group, defined as a macular inward convexity in some meridians across the fovea, whereas the opposite perpendicularly oriented meridians were flat. Age, spherical equivalent, axial length (AL), and best-corrected visual acuity (BCVA) were collected. Height of the bulge, scleral and choroidal thicknesses, Bruch´s membrane defects, and presence of perforating scleral vessels were recorded. Results: Thirty-seven (75.5%) eyes were classified into the RSM group and 12 (24.5%) into the DSM group. Twenty-six (53.0%) eyes showed macular elevation only in the horizontal direction. Mean AL showed statistically significant differences (28.8 ± 2.7 vs. 30.5 ± 1.5 mm in the RMS vs. DSM group, respectively) and the presence of Bruch´s membrane defects was more frequently seen in DSM (*p* < 0.001). Mean age, spherical equivalent, BCVA, height of the inward convexity, retinal foveal thickness, foveal scleral thickness, subfoveal choroidal thickness, and the presence of perforating scleral vessels did not show significant differences between groups. Conclusion: This study shows the reliability of using the 12 equal radial OCT scans as an objective method to define and differentiate DSM versus RSM. Patients with RSM showed differences in AL compared with those with DSM, being longer in DSM, and regarding the presence of Bruch´s membrane defects, being more common in DSM. This may contribute to identifying those patients that, in daily clinical practice, have a higher risk of developing complications due to their myopia.

## 1. Introduction

Pathologic myopia is one of the leading causes of legal blindness in developed countries. Axial elongation of the globe leads to complex changes in the topography of the posterior pole, with thinning of the retina, choroid, and sclera and subsequent development of macular pathologic features [1].

Advances in optical coherence tomography (OCT), most notably seen in swept source (SS), which uses a longer central wavelength, generally in the 1 µm range, offer deep penetration imaging. This technology has improved the visualization of deeper structures and has provided good quality images of choroid, sclera, retrobulbar fat, and even vascular structures of the posterior pole [2,3].

Dome-shaped macula (DSM) was first described by Gaucher in 2008 as an inward protrusion of the macula, as visualized by OCT [4]. Although it was first described as a dome-shaped macula, later, using 3D image reconstruction of the posterior pole topography by SS-OCT, Ellaban et al. [5] reported the precise topography, describing a horizontal ridge formed within the posterior staphyloma. Caillaux and colleagues [6], using spectral domain (SD) OCT, described three morphologic patterns according to tomographic features: typical dome-shaped convexity defined as round domes (without predominant axis), more common horizontally oriented band-shaped domes traversing the posterior staphyloma, and rare vertically oriented oval-shaped domes.

Many other definitions have been used by other authors, such as convexity of the retina-choroidal macular complex on OCT scans [7] or bidirectional dome, seen in horizontal and vertical OCT sections [8], with recognition in the literature that this macular curvature is not a rare finding in highly myopic eyes, with an estimated prevalence of 10.7% in a European study and 9.3% in a Japanese study [5,9].

Recently, ridge-shaped macula (RSM) has been the term used to define the macular elevation only in one meridian across the fovea, whereas the macular curvature was unremarkable in the perpendicular meridian, different from typical bidirectional domes [8].

The aim of our study was to use the OCT as a method for classifying these patients and to analyze the clinical features of these two different types of macular bulges in order to better understand the characteristics of these posterior pole alterations, starting from the concept of a ridge as an elevation in one orientation without elevation of its perpendicular versus a round dome as a complete inward convexity of the macular area (Figure 1).

## 2. Materials and Methods

The procedures used in this retrospective observational study adhered to the tenets of the Declaration of Helsinki and its protocol was approved by the Ethics Committee of Puerta de Hierro-Majadahonda University Hospital (Spain) with the approval code PI 91/18. The case series study included 49 highly myopic eyes from 31 patients, visited between January 2019 and January 2020 with a macular convexity in SS-OCT. 

Patient inclusion criteria comprised the presence of high myopia defined by a myopic refractive error (spherical equivalent) of ≥6.0 diopters(D) or an axial length ≥26 mm and a macular convex elevation on the posterior pole defined as an inward bulge of the retinal pigment epithelium (RPE) and Bruch´s membrane (BM) with a height ≥50 µm, according to the definition of Ellabban et al. [5] and Ohsugi et al. [9].

The exclusion criteria for the recruitment included poor quality OCT images (<45 image quality score of DRI Triton SS OCT software), and previous vitreoretinal surgeries since this could have affected the scleral curvature. Eyes with the presence of other retinal diseases such as retinal vascular disorders, macular dystrophies or age-related macular degeneration, history of ocular trauma, uveitis, or glaucoma were excluded as well.

All study participants underwent a comprehensive demographic interview and a complete ocular examination including gender, age, refractive error determination (spherical equivalent) with an autorefractometer (Accuref-K 9001, Shin-Nippon, Tokyo, Japan), axial length (AL) measurement using optical biometry (IOL Master 500, CarlZeiss, Germany), and best-corrected visual acuity (BCVA).

Color fundus photography and SS-OCT images were taken using a Topcon DRI Triton SS OCT (Topcon Corp., Tokyo, Japan), performing 9 mm 12 radial equal meridian scans centered on the fovea and containing 1024 axial scans. A 3D macula image was also acquired using a raster scan of 512 × 128 A-scans covering an area of 7 mm^2^ centered on the fovea. Height and orientation of the inward convexity, retinal and scleral foveal thicknesses, and choroidal thicknesses at the fovea and at four parafoveal locations 1500 µm from the foveal center were measured. Bruch´s membrane defects and the presence of perforating scleral vessels were recorded.

The classification in both groups (DSM and RSM) was made based on objective criteria, considering the number of affected OCT radial scans centered on the fovea. DSM eyes were defined as those that showed a complete round inward convexity in all their axes, presenting an inward convexity ≥50 µm in the 12-line radial OCT scans. Eyes that did not meet this criterion and had at least one flat radial scan were grouped into the RSM group. Three retinal specialists classified the patients in both groups in a masked fashion (M.G.Z., I.F.M., and J.R.M.). Retinal, choroidal, and scleral measurements were manually performed.

The height of the macular bulge was measured as the distance between the highest point of the bulge and a line running parallel to the RPE at both sides of the base of the bulge on the OCT section showing the maximal height. Retinal thickness was defined as the distance between the inner retinal surface and the outer border of the RPE. To measure the choroidal thickness, the distance between BM and the outer part of the hyperreflective margin line corresponding to the choroid–scleral interface was measured [10]. Scleral thickness was established as the distance between the choroid–scleral interface and the end of the hyperreflective scleral image. Retinal, subfoveal choroidal, and scleral thicknesses were determined in the vertical OCT section across the fovea. The presence of BM defects around the macular elevation [11] and the occurrence of penetrating scleral vessels as reported by Ohno-Matsui et al. were analyzed [12].

### Statistical Analysis

Descriptive statistics were provided using mean ± SD (standard deviation) for quantitative and n(%) for categorical variables. For statistical treatment of the data, SPSS for Windows (SPSS, Chicago, IL) was used (version 24.0). The significance of the differences in age; AL; maximal dome height; and retinal, choroidal, and scleral thicknesses between the groups was determined using the unpaired Student’s *t*-test. Differences in the presence of BM defects or penetrating scleral vessels between the groups were analyzed by applying Fisher´s exact test. A *p*-value < 0.05 was considered statistically significant.

## 3. Results

During the study period, 349 highly myopic patients were examined, out of which 49 eyes (14.0%) of 31 patients showed macular elevation that met the definition of an inward convexity of RPE and BM ≥ 50 µm.

A typical round dome-shaped macula, with the 12 radial OCT scans centered on the fovea showing an elevation of ≥50 µm, was found in 12 eyes (24.4%), defining the DSM group (Figure 2). However, despite being round, all of them presented a predominant orientation, with some scans displaying a higher bulge than the opposing scans. Nine eyes of the DSM group showed a predominant horizontal orientation (75%) versus three eyes that showed a higher bulge in the vertical orientation (25%).

In the RSM group, we included those 37 eyes (75.6%) with <12 OCT radial scans affected. The mean of the OCT scans that showed an inward elevation ≥50 µm was 5.75 ± 1.69. Twenty-six eyes (70.2%) showed only an inward convexity in the horizontal direction manifested across the vertical OCT radial scans, whereas the opposite scans were flat, being classified as horizontally oriented RSM (Figure 3A). Ten (27.0%) eyes showed this inward convexity only in the vertical direction across the horizontal OCT radial scans, fulfilling the definition of a vertically oriented RSM (Figure 3B). One eye (2.7%) was classified as obliquely oriented RSM. The inward convexity occurred bilaterally in 17 eyes (34.6%). In four patients (8.1%), one eye showed the typical round dome-shaped convexity and the other eye showed a ridge-shaped band.

A statistical comparison between the RSM and DSM group showed no differences in mean age (65.1 ± 12.0 vs. 59.1 ± 14.1 years in RMS vs. DSM group respectively, Student’s *t*-test, *p* = 0.15), spherical equivalent (−12.2 ± 7.6 vs. −17.1 ± 3.7, Student’s *t*-test, *p* = 0.14), logMAR BCVA (0.57 ± 0.58 vs. 0.30 ± 0.21, Student’s *t*-test, *p* = 0.12), height of the inward convexity (272.7 ± 165.3 vs. 329.6 ± 194.8 µm, Student’s *t*-test, *p* = 0.32), retinal foveal thickness (201.3 ± 64.8 vs. 195.9 ± 68.1 µm, Student’s *t*-test, *p* = 0.80), foveal scleral thickness (335.1 ± 115.5 vs. 368.0 ± 136.0 µm, Student’s *t*-test, *p* = 0.41), subfoveal choroidal thickness (116.9 ± 78.3 vs. 103.1 ± 52.3 µm, Student’s *t*-test, *p* = 0.57), and the presence of CNV (Fisher test, *p* = 0.73) or perforating scleral vessels (Fisher test, *p* = 0.31) (Table 1).

Mean axial length showed statistically significant differences, being longer in DSM eyes (28.8 ± 2.7 mm vs. 30.5 ± 1.5 in the RMS vs. DSM group, respectively, Student’s *t*-test, *p* = 0.014).

Within the group of eyes with DSM, one or more BM defect was found in 9 of the 12 eyes classified herein (75%), while in the RSM group, the presence of BM defect was found in 5 of the 37 eyes (13.5%), with the prevalence of BM defects being significantly higher in the DSM group (Fisher test, *p* < 0.001). Among BM defects (14 eyes), 3 patterns were differentiated: a patchy atrophy pattern (35.7%), BM defects associated with CNV-related macular atrophy (14.2%), and BM defects as an extension of a peripapillary atrophy (35.7%). One eye (7.1%) showed two different patterns associated with the BM defects (patchy atrophy with a CNV related macular atrophy) and one eye (7.1%) showed a patchy and peripapillary atrophy (Figure 4).

Among the concomitances of the entire sample of 49 eyes, choroidal neovascularization (CNV) was seen in 15 eyes (30.61%), serous retinal detachment in 6 eyes (12.2%), lamellar macular hole in 3 eyes (6.1%), epiretinal membrane in 1 eye (2.0%), and macular schisis in 2 eyes (4.0%).

## 4. Discussion

This study establishes an objective method of classification of the macular inward convexities based on the number of 12-line fovea-centered radial OCT scans affected. DSM was considered when all 12 radial scans showed a macular elevation of RPE and BM line ≥50 µm. RSM was defined when at least 1 of the 12 radial scans was not affected and showed a flat profile, with the mean radial scans being affected in near 6 out of 12. This may suggest that most of the RSM cohort showed an inward convexity in one of the perpendicular meridians represented by the six OCT radial scans oriented in that direction, whereas the opposite radial scans were flat or almost flat.

Gaucher et al. described for the first time this posterior pole alteration in myopic eyes, which they named DSM, performing cross-sectional images with mapping software and multiple line software with 4, 5, 6, or 8 mm long scans. One eye was examined with SD-OCT (3D OCT-1000; Topcon, Tokyo, Japan), which allowed three-dimensional (3D) reconstruction of the elevated macula in the staphyloma [4]. Later, Caillaux et al., using horizontal and vertical SD-OCT scans and 3D macular map reconstruction, described three morphologic DSM patterns: type one: round domes; type two: horizontally oriented oval-shaped domes, splitting the staphyloma into two parts, superior and inferior; and type three: vertically oriented oval-shaped domes, splitting the staphyloma into two parts, temporal and nasal [6]. Our classification would be similar to these three types of macular bulges described by the authors, considering the term ´ridge´ a better descriptive term for those vertically or horizontally oriented domes that simulate a crest on the posterior pole. At the same time, we incorporate an objective classification measure, basing our results on the number of 9 mm 12 equal radial OCT scans affected, which provided us with similar percentages as in previous studies [5,6,8,9]. Ellaban et al. differentiated the typical round dome from another type of macular convexity named the band-shaped ridge macula as a result of two outward concavities in the posterior staphyloma with a horizontal ridge between them, shown in a 3D image reconstruction [5]. In each case, SS-OCT horizontal and vertical scans of 12 mm were obtained and 3D imaging data sets were acquired. Ohsugi et al. measured the bulge height also based on the vertical and horizontal OCT sections classifying the DSM into three groups: a round dome group (both horizontal and vertical sections affected), a horizontally oriented oval-shaped dome group (vertical section was dome-shaped, while horizontal section was almost flat), and a vertically oriented oval-shaped dome group (horizontal section was dome-shaped, while vertical section was almost flat) [9]. Fang et al. [11] based their results on SS-OCT images, including 6 mm or 9 mm length of radial scans with 12 equal meridian scans centered on the fovea, as performed in the current study. DSM was defined as an inward bulging of the RPE line with a height ≥ 50 µm above the base RPE line, but the number of radial sectors that define a complete dome or a horizontally or vertically oriented dome was not specified. 

Recently, RSM has been the term used by Xu et al. [8] to describe eyes in young patients that showed a particular elevation of the macula only in one meridian with an aspect of a ridge, differentiating it from a DSM, which shows an elevation in all meridians with a round configuration in all directions. This definition was based on SS-OCT horizontal and vertical raster scans of 12 mm, and the elevation of the macula was defined as an inward bulge of RPE and BM of ≥50 µm on the vertical OCT sections.

In the current study, a comparison of features was performed between patients with DSM and an extended definition of RSM, considering as ridges those macular elevations that, owing to the shaped they create in the posterior pole, showed only elevation in one meridian, with this group being previously defined by other authors as DSM horizontally or vertically oriented oval-shaped maculas [6]. We based this consideration on anatomical characteristics of the posterior pole, taking into account that patients with RSM showed lower height of the macular elevation and significantly less association with BM defects.

The comparison between groups showed that mean age, spherical equivalent, BCVA logMAR, height of the inward convexity, retinal foveal thickness, foveal scleral thickness, subfoveal choroidal thickness, and the presence of perforating scleral vessels did not show significant differences between groups. In the literature, the mean age of patients with a DSM usually ranged between 50 and 60 years [4,5,8,9]. However, Xu et al. reported in the comparison they made between RSM and DSM eyes that this was last detected only in patients older than 20 years, with RSM prevailing in those under 20 years of age [8]. In our study, significant differences were not found, with a similar mean age in both groups, with the limitation that, in our cohort, patients below 20 years old were not included.

The height of the macular elevation was lower in the RSM group than in the DSM group, but this difference did not reach statistical significance in our cohort. The mechanisms under this anatomic elevation remain unknown, while different theories have been postulated, with it remaining unclear if a bidirectional dome and a unique directional dome (RSM) may have a common etiology.

Imamura et al. reported that a DSM could be potentially caused by a focal thickening of the subfoveal sclera, postulating that this thickening was an adaptative response to the image defocus on the fovea of highly myopic patients [12]. As in other studies [5,8], subfoveal choroidal thickness was thinner in DSM than in RSM, and scleral thickness was thicker in DSM than in RSM, but it was not statistically different in our cohort. Axial elongation in the process of myopization is associated with thinning of choroidal and scleral thicknesses, mostly posterior to the equator in the posterior pole [12,13,14].

Although many previous authors suggested that the sclera and/or choroid might be primarily responsible for eye elongation, recent studies have discussed the role of the BM as a main factor in the process [11,15,16]. Jonas et al. [17,18] described that retinal thickness and RPE density in the macular region and thickness of BM in any region were not related to axial length, suggesting the hypothesis that axial elongation occurs by production of BM in the retro-equatorial region, leading to a decreased RPE density and retinal thinning in that region [16,17,18,19]. 

Fang et al. observed that the presence of a DSM was significantly associated with the presence of macular BM defects, assuming that these defects may allow a local relaxation of the posterior sclera that would then be no longer be pushed outward, developing a bulge inward [11]. Based on the previously stated theory, BM may at the same time produce an expansion from the retro-equatorial region posteriorly, compressing the posterior choroid and pushing the posterior sclera, adding to the effect of the biomechanical hypothesis [12,13,14]. As BM may have these biomechanical properties in terms of strength, the defects found around the macula may release the central island from biomechanical forces so that it can slightly bulge inward, forming the dome.

In our study, the presence of BM defects around the dome showed statistically significant differences between groups, with a higher prevalence in the DSM group (Fisher test, *p* < 0.001), in the same way as the AL, with these DSM eyes being significantly longer (*p* = 0.01). Based on the findings described in other studies, AL is the main variable associated with myopic maculopathy. Flores et al. showed that patients with severe pathologic myopia (PM) have longer AL compared with a PM group, being positively correlated with the A, T, and N components of the ATN grading system [20]. Other authors also previously showed the correlation between AL and the atrophic component of myopic maculopathy, indicating the clinical importance of AL in detecting PM [20,21].

Differentiating these patients with DSM and greater extensions of BM defects or longer AL could help us in daily clinical practice to identify those high myopic patients with a greater risk of progression towards degrees of severe pathologic myopia. In this way, professionals could carry out more exhaustive follow-up to these patients versus those with RSM, without BM defects, or with shorter axial lengths.

The limitations of our study should be included. The first is the sample size. Despite being a rather large series of domes, the number of subjects included in the sample is small for statistical analysis. Second, we have the limitation of not having much data from the young population. Third, the hypothesis of BM having an active participation in the process of emmetropization and myopization is still unproven, despite the high number of recent publications [11,15,16,17,18]. Finally, a more exhaustive analysis wherein we could include the presence or not of posterior staphyloma could reveal more data on the etiopathogenesis of these entities.

## 5. Conclusions

This study proposes to extend and consider the use of the term “ridge” for those oval-shaped domes previously defined in the literature that do not show a complete round elevation in all meridians, whether horizontally or vertically oriented, and regardless of age. At the same time, it shows the reliability of using the 12-line equal radial OCT scans as an objective method to define and differentiate DSM versus RSM. Patients with DSM versus those with RSM showed a statistically significant difference regarding AL, that is, more elongated eyes, and the presence of BM defects, being more frequently seen in the DSM group, allowing professionals to identify those patients with a higher risk of progression to pathologic myopia.

## Figures and Tables

**Figure 1 diagnostics-11-01864-f001:**
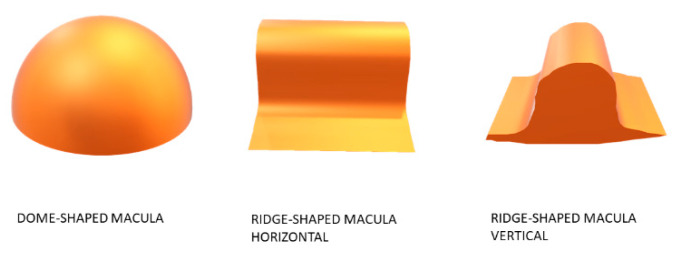
Geometric concepts of dome and ridge.

**Figure 2 diagnostics-11-01864-f002:**
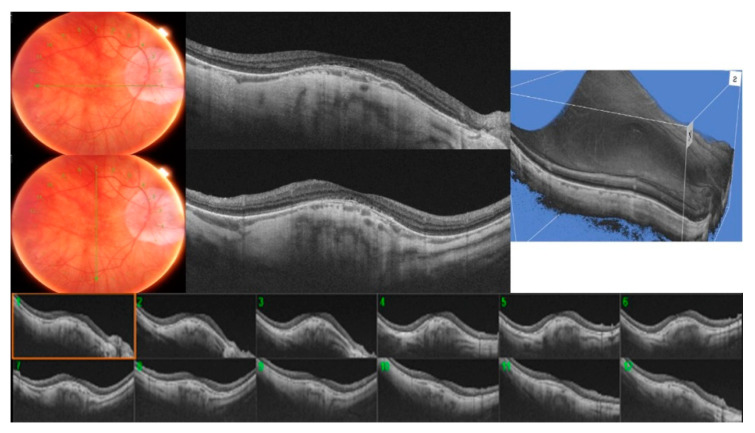
A typical round dome-shaped macula (DSM) with the 12 radial OCT scans centered on the fovea showing an inward convexity of ≥50 µm.

**Figure 3 diagnostics-11-01864-f003:**
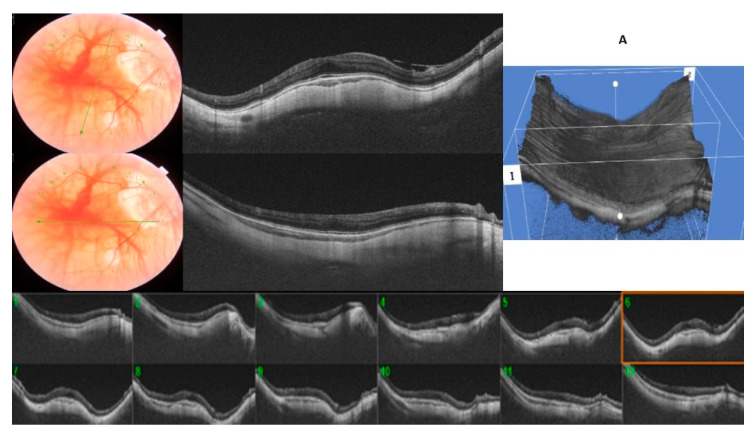
(**A**)/(**B**): A ridge-shaped macula (RSM) with six radial OCT scans centered on the fovea showing an inward convexity of ≥50 µm, whereas the macular curvature was unremarkable on opposite perpendicular meridians. (**A**): horizontally oriented RSM. (**B**): vertically oriented RSM.

**Figure 4 diagnostics-11-01864-f004:**
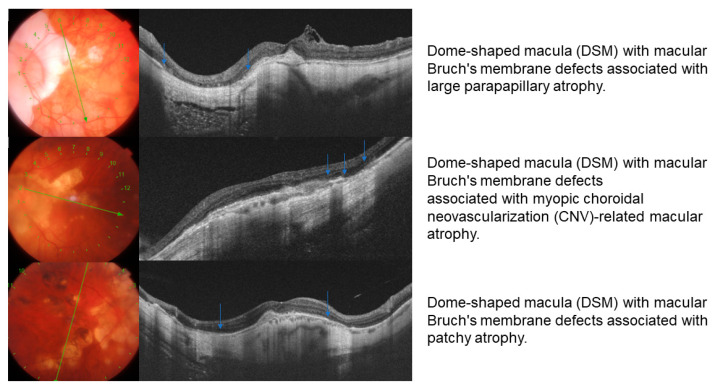
Bruch membrane defects: different patterns. Blue arrows mark the areas with Bruch´s membrane defects.

**Table 1 diagnostics-11-01864-t001:** Comparison data between the DSM and RSM groups.

	DMS	RSM	*p*-Value
No. of eyes	12 (24.5%)	37 (75.5%)	-
Age (years)	59.17 ± 14.18	65.19 ± 12.08	0.15
Female eyes (n, %)	7 (58.3)	23 (62.1%)	0.53
Axial length (mm)	30.57 ± 1.54	28.89 ± 2.78	0.01
Spherical equivalent (D)	−17.16 ± 3.71	−12.23 ± 7.66	0.14
BCVA (logMAR)	0.30 ± 0.21	0.57 ± 0.58	0.12
High (µm) macular convexity	329.67 ± 194.88	272.76 ± 165.33	0.32
Orientation horizontal (n, %)	9 (75.0%)	26 (70.27%)	0.53
Retinal foveal thickness (µm)	195.98 ± 68.15	201.32 ± 64.89	0.80
Foveal scleral thickness (µm)	368.08 ± 136.01	335.16 ± 115.55	0.41
Subfoveal choroidal thickness (µm)	103.17 ± 52.31	116.95 ± 78.32	0.57
Choroidal 1500 temporal thickness (µm)	69.17 ± 34.00	94.70 ± 68.57	0.22
Choroidal 1500 nasal thickness (µm)	54.00 ± 44.28	75.76 ± 56.59	0.23
Choroidal 1500 superior (µm)	74.00 ± 42.60	105.59 ± 83.97	0.21
Choroidal 1500 inferior thickness (µm)	74.75 ± 35.35	80.49 ± 52.46	0.72
BM defects (n, %)	9 (75%)	5 (13.51%)	0.000
Perforating scleral vessels (n, %)	12 (100%)	31 (83.7%)	0.16
CNV (n, %)	3 (25%)	12 (32.4%)	0.73
Serous RD (n, %)	3 (25%)	3 (8.10%)	0.14

## Data Availability

All data generated or analysed during this study are included in this article. Further enquiries can be directed to the corresponding author.

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
