# Peer review of "Dome-Shaped Macula versus Ridge-Shaped Macula Eyes in High Myopia Based on the 12-line Radial Optical Coherence Tomography Scan Pattern. Differences in Clinical Features"

_diagnostics, 2021, doi:10.3390/diagnostics11101864_

Round 1

Reviewer 1 Report

I think is a very relevant topic for daily clinical practice.
While it is true that the conclusions do not make clear how it can help the professional on a day-to-day basis, it indicates new advances in the classification.
The structure of the article is adequate and the applied methodology has been well developed.
The results obtained are described to perfection, but a better explanation should be attempted for a better understanding.
In relation to the discussion, the authors shows a deep compare with other authors.
I miss how you can help the professional in the daily clinic.
The conclusions should be clearer and avoid repeating results.
I don't consider it necessary as conclusions to say what can be done in the future. It is true that the investigation must open new lines but if these are too many, it means that it is not complete

Author Response

First of all, thank you very much for your contributions to the article. Your comments and ideas improve the quality of it and we are very grateful to consider all of them.

  • Regarding the contribution of the article to daily clinical practice, we have another work recently published by our group where we show that AL is the main variable associated with myopic maculopathy. Patients classifed with severe pathologic myopia (PM) had longer AL and worse BCVA (p<0.05) compared to the PM group. In that study, axial length was positively correlated with the A, T and N components of the ATN grading system. These results are consistent with previous studies that assessed the correlations between AL and the atrophic components of myopic maculopathy. Xiao et al. found that the odds of developing myopic atrophic maculopathy (multivariate model) was 2.97 times higher for each millimetre increase in AL. From our results, signifcant betweengroup diferences in AL according to the A grade (ATN system) were observed, as follows: A2 (difuse atrophy), 2.17 mm; A3 (patchy atrophy), 3.36 mm; and A4 (complete macular atrophy), 3.47 mm. Those findings reveal significant diferences in AL among eyes considered PM (≥ A2), thus indicating the clinical importance of AL in detecting PM. (Flores-Moreno I, Puertas M, Almazán-Alonso E, Ruiz-Medrano J, García-Zamora M, Vega-González R, Ruiz-Moreno JM. Pathologic myopia and severe pathologic myopia: correlation with axial length. Graefes Arch Clin Exp Ophthalmol. 2021 Aug 18. doi: 10.1007/s00417-021-05372-0). The fact of identifying greater axial lengths among patients with dome-shaped macula could help to identify on a daily basis these patients that have a greater risk of complications related to their myopia. We have added these data to the article thanks to your contribution. Very thankful.
  • We have tried to clarify and specify the conclusions to synthesize the idea that we want to convey. Thank you very much for the recommendation.

Sincerely, 

Maria Garcia 

Reviewer 2 Report

This is an overall well written and interesting paper comparing highly myopic eyes with dome-shaped (DSM) and ridge-shaped (RSM) macula.

Since the use of 12 radial SS-OCT B-scans has been previously used to determine the axial extension of the inward convexity of the posterior buldge in these eyes, I would suggest to avoid the term "classification" in the manuscript title. In fact, the results of the present study seem to validate this OCT pattern as a useful method to distinguish DSM and RSM, rather than providing a novel classification scheme, as vaguely evoked by the title.

It would be also interesting to know the association of DSM and RSM with posterior staphylomata in the present series, as evaluable by ophthalmoscopic and OCT appearence of these eyes. In their 2018 paper, Ohno Matsui et al. (Xu X, Fang Y, Jonas JB, Du R, Shinohara K, Tanaka N, Yokoi T, Onishi Y, Uramoto K, Kamoi K, Yoshida T, Ohno-Matsui K. RIDGE-SHAPED MACULA IN YOUNG MYOPIC PATIENTS AND ITS DIFFERENTIATION FROM TYPICAL DOME-SHAPED MACULA IN ELDERLY MYOPIC PATIENTS. Retina. 2020 Feb;40(2):225-232) found that young myopic subjects had RSM not associated with staphylomata, hypothetically due to Bruch's membrane folding at the posterior pole. These authors wonder if this population is prone to develop staphylomata sorrounding the RSM later in life, and also other researchers tried to establish if there is a relationship between DSM and posterior sthaphyloma (Dai F, Li S, Wang Y, Li S, Han J, Li M, Zhang Z, Jin X, Dou S. CORRELATION BETWEEN POSTERIOR STAPHYLOMA AND DOME-SHAPED MACULA IN HIGH MYOPIC EYES. Retina. 2020 Nov;40(11):2119-2126. doi: 10.1097/IAE.0000000000002722.). This could be a good occasion to provide some data about this.

Author Response

First of all, thank you very much for your contributions to the article. Your comments and ideas improve the quality of it and we are very grateful to consider all of them.

  • The suggestion to change the title of the paper and avoid the term “classification”, seems to us a totally very good idea, indicating just that this OCT pattern is a useful method to distinguish DSM and RSM. Thank you very much for the recommendation.
  • Regarding the consideration of posterior staphyloma in this article, it is a brilliant idea nowadays due to the great recent articles on this topic. It could not be one of the objectives of our work because to obtain those data we considered necessary to have a wide-field OCT or a magnetic resonance imaging (MRI) as so do other groups and in this way be able to compete in quality with these latest published works. In our center, we still do not have any of them for the study of myopic pathology and it seemed to us that reviewing posterior staphylomas only by fundus examination and OCT- B scans could be misleading. If you still consider it, we would review the patients with some time needed, classifying their posterior staphyloma with the means that we have. If not, we would of course consider it for a future extension of the work. Thank you very much for the recommendation that would really improve our work.

Sincerely,

Maria Garcia
